



# Oxygen dynamics and evaluation of the single station diel oxygen model across contrasting geologies

Simon J. Parker[1]

[1]Teagasc, Johnstown Castle, County Wexford, Ireland.

**Correspondence:** Simon Parker (simon.bierton@gmail.com)

**Abstract.**

In aquatic ecosystems, the single station diel oxygen model assumes constant ecosystem respiration and aeration rate (notwithstanding temperature effects) over the course of a single night. This assumption was tested for four small streams representing two geologies (Chalk and Greensand) over a one year period, by examining the behaviour

of the night-time dissolved oxygen (DO) saturation deficit at points where change in DO is zero. This was then used as a proxy for the ratio of aerobic ecosystem respiration (R) to the aeration rate constant (k) and compared with the corresponding ratio (the regression quotient) obtained from night-time regression analysis (Hornberger and Kelly, 1975). For two streams (one Chalk and one Greensand), the regression quotient persistently underestimated the observed DO deficit. These two streams showed similar timing patterns of oxygen dynamics with the point of

minimum DO occurring relatively quickly after sunset in spring and early summer, although the two Chalk streams were more similar to one another in terms of DO magnitudes. Therefore, characterisation of a Chalk stream is not just dependent on geology. Comparisons between different streams using the single station model on the presumption that it is equally appropriate in all cases may lead to misleading conclusions.

**Introduction**

The dissolved oxygen (DO) signal has been used to quantify primary productivity and respiration in aquatic ecosystems since the pioneering work of Odum (1956). Recently, the increased capacity to deploy automatic data loggers coupled with the ability to automate the analysis of the DO signal (e.g. Grace *et al.*, 2015) has enabled the processing of potentially large amounts of data across multiple aquatic systems. Estimates of primary production obtained from the DO signal can then be used through the photosynthetic quotient (e.g. Duarte *et al.*, 2010; Westlake, 1963) to

estimate the corresponding carbon uptake. Therefore, with growing awareness of the significance of river systems in





global carbon cycling (Cole *et al.*, 2007; Wohl *et al.*, 2017) it becomes more relevant to ensure both that the models used are sound and also that model limitations are apparent.

Primary production can be quantified by partitioning a single DO time series into its component fluxes, namely photosynthesis, ecosystem respiration and aeration. Although for certain components of aquatic systems, oxygen

consumption can be measured continuously, for example, through the use of benthic incubation chambers (e.g. Glud, 2008) or using eddy correlation techniques (e.g. Reimers *et al.*, 2012), there is no method to measure oxygen consumption for the whole system. For aeration, although it is possible to measure the gas exchange constant using tracers such as sulphur hexafluoride (e.g. Beaulieu *et al.*, 2013) or propane (e.g. Demars *et al.*, 2011), from which the exchange constant for oxygen can be derived, only recently has a method been proposed (Pennington *et al.*, 2018)

to do this on a continuous basis. This means that time series estimates of oxygen consumption for a whole stream are coupled to estimates of the aeration flux and must be inferred, rather than measured, from DO time series, so that quantification of each depends on simultaneously quantifying the other.

In order to make such inferences, both community respiration (R) and the volumetric aeration rate constant (k) are typically assumed to be constant (notwithstanding temperature effects) over one diurnal cycle (e.g. Correa-Gonzalez

*et al.*, 2014; Izagirre *et al.*, 2008; Benjamin *et al.*, 2016; Richmond *et al.*, 2016). Yet, the assumption of constant respiration is questioned (Staehr *et al.*, 2010) and recent research (Sadro *et al.*, 2014; Schindler *et al.*, 2017) suggests that R is better represented by a two stage process according to whether the carbon source is autochthonous or allochthonous and there is experimental evidence that dark respiration rates can decrease as the night progresses (Alnoee *et al.*, 2014). A further obstacle with single stage R models is that identification of R and k is hindered

by equifinality (Appling *et al.*, 2018; Beven, 2006) which is that multiple pairs of (R, k) values can equally well explain a particular DO times series. Together these pose substantial obstacles in the quantification of whole-stream metabolism.

The open channel diel method requires the partitioning of the stream dissolved oxygen response into the dominant processes as described by the following equation (disregarding temperature effects as they are not the focus of this

research):

$$\frac{d(DO)}{dt} = P - R + k(DO_{sat} - DO)$$

where

DO is dissolved oxygen concentration (g $O_2$ m$^{-3}$)

P is the oxygen flux resulting from photosynthesis (g $O_2$ m$^{-3}$ s$^{-1}$)

$-$R is the oxygen consumption resulting from aerobic respiration (g $O_2$ m$^{-3}$ s$^{-1}$)

$DO_{sat}$ is dissolved oxygen concentration at saturation (g $O_2$ m$^{-3}$)

t is the time (seconds)

k is volumetric aeration rate constant (s$^{-1}$)



For night-time, this relationship simplifies to:

$$\frac{d(DO)}{dt} = -R + k(DO_{sat} - DO)$$

Therefore, when $\frac{d(DO)}{dt} = 0$

$$\frac{R}{k} = (DO_{sat} - DO)$$

Therefore, assuming this model correctly describes the pertinent processes, at points of zero DO change in the night-time DO time series the ratio of respiration to the volumetric aeration rate constant is equal to the observed

oxygen saturation deficit. Thus, by identifying points in time of zero DO change, ($DO_{sat}$ - DO) can be observed from which the ratio $\frac{R}{k}$ can be inferred. These observations can then be compared with theoretical counterparts by using the night-time regression method (Hornberger and Kelly, 1975) to obtain values of respiration ($R_{HK}$) and k ($k_{HK}$) and by extension the quotient ($\frac{R_{HK}}{k_{HK}}$), hereafter referred to as the regression quotient.

The questions addressed are:

(1) How does the observed oxygen saturation deficit at points of zero DO change ($DOD_{zero\ \Delta DO}$) behave over time?

(2) How do night-time $DOD_{zero\ \Delta DO}$ values (as proxies for $\frac{R}{k}$) compare with the regression quotient ($\frac{R_{HK}}{k_{HK}}$)?

(3) Does the time at which $DOD_{zero\ \Delta DO}$ occurs depend on the underlying stream geology?

## Methods

### Study area

The study was conducted in the southern part of the United Kingdom in the Hampshire Avon catchment. The catchment covers an area of 1706 $km^2$ (NFRA, 2018) and has an average annual rainfall of 810 mm. Approximately 80% of the catchment is arable or grassland and less than 2% is urban. The dominant geology in the catchment is highly permeable Chalk so that the rivers are primarily groundwater fed. Instrumentation was located on four

tributaries within that catchment, the rivers Ebble, Wylye, Nadder and Upper Avon (Table 1) with surface water catchment sizes between 35 and 59 km$^2$ and two dominant geology types (Chalk and Greensand). A more detailed site description is available in Heppell *et al.*, 2017.

### Instrumentation and data analysis

Dissolved oxygen and temperature were logged continuously using miniDOT data loggers (Precision Measurement

Engineering, Inc.) at a resolution of 0.01 mg per litre and logging frequency of 1 minute from mid-August 2014 to mid-August 2015. The DO time series for the miniDOTs was smoothed using a 30 minute time step with the change



in DO ($\Delta$DO) at each minute computed from the smoothed time series. From this, the time at which $\Delta$DO = 0 was identified and the associated value of the DO deficit was noted. Dissolved oxygen at saturation was calculated using tables provided by United States Geological Survey (USGS, 2015) in accordance with Standard Methods of the American Public Health Association (1998), using both water temperature and atmospheric pressure, with

atmospheric pressure data provided by British Atmospheric Data Centre. For the night-time regression calculation, those data points that incorporated daytime values as a consequence of the implemented moving average were excluded from the regression. The data reported in this study is available from the NERC data centre (Heppell and Parker, 2018).

**Results**

Figure 1 shows the DO time series (raw data) for a two week period in May 2015. For the two Chalk rivers, daytime DO consistently rises above $DO_{sat}$ typically by 1 to 3 mg DO per litre for the Wylye and 1 to 2 mg DO per litre for the Ebble. For the Greensand rivers, the Nadder rarely rises above saturation and although the Avon does so, nevertheless not as regularly as the two Chalk rivers. The Avon shows anomalous behaviour for the 13th and 14th May. Average daily DO maxima are 12.7, 12.2, 11.7 and 10.9 mg DO per litre for the Wylye, Ebble, Avon and Nadder

respectively, so that *prima facie* the Wylye is the most productive. Peak daytime DO for the Wylye tends to happen later than that for the Ebble, as does the peak for the Avon compared to the Nadder, so that for example in the daytime of May 16th, DO for the Ebble and Wylye rises to 12 mg per litre, after which DO in the Ebble declines whilst DO in the Wylye continues to rise to 13.5 mg DO per litre. Note also that for the Ebble night-time DO reaches a minimum early each night, after which it rises throughout the night, whereas the Wylye shows two types of

behaviour, so that for example on the nights of 13th/14th and 16th/17th May, minimum DO occurs early whereas for 6th/7th and 9th/10th minimum DO occurs much later in the night. This behaviour is summarised in Figure 2, where panel A shows DO distributions and panel B, the DO concentration averaged by time after sunrise (Figure 2(B)), which shows that DO concentrations for the Ebble and Nadder typically plateau at solar noon, whereas those for the Wylye and Avon continue to rise until 2 to 4 hours after solar noon.

In fact, the behaviour of the Ebble in terms of timing (i.e. phase) is much closer to that of the Nadder than to the behaviour of the Wylye. Figure 3 (panels A and B) shows the distributions of differences in DO at different lag intervals between de-trended (that is, mean DO is first subtracted) DO time series for the Wylye, Ebble and Nadder. Each boxplot is the distribution of the difference in DO for two of those time series, with one time series having been time-shifted by the number of minutes shown on the x-axis. For the two Chalk streams (Ebble and Wylye, panel

30   A), the Wylye tends to respond later than the Ebble and is phase-shifted by approximately 90 minutes, whereas for the Ebble and Nadder (panel B), both systems respond at approximately the same time. The cross-correlations in Figure 3C summarise the relative timings for all four rivers; the correlation is stronger for the Ebble and Wylye and for the Ebble and Nadder than it is for the Wylye compared to the Avon and also the Nadder compared to the Avon





(for which anomalous data of 13th and 14th May was removed prior to analysis). Nevertheless, for the whole time series, the Avon lags the Nadder by approximately two hours, which is consistent with Figure 2. Thus, in terms of typical DO magnitudes, the two Chalk streams are similar (Figure 2A), but in terms of phase the Ebble is similar to the Nadder.

For the time series shown in Figure 1, the nights of May $9^{th}$ to May $10^{th}$ and of May $16^{th}$ to May $17^{th}$ are shown as examples in Figures 4 and 5 showing both raw DO data (grey circles) and a 30 point DO moving average (solid black line), together with associated changes in DO at each minute. The changes in DO are computed using the 30 point DO moving average, not the raw data. Black triangles are those points where $\Delta DO = 0$, discussed further below. At sunset, the Wylye shows the greatest rate of DO decline of -0.016 g $O_2$ $m^{-3}$ per minute with the Ebble

and the Nadder each experiencing approximately half that rate, with the Nadder considerably less on the 9th/10th May. For the Avon, the initial rate of decline is intermediate between those at about -0.010 g $O_2$ $m^{-3}$ per minute. For all four rivers, the rate of decline at sunset is higher on the 9th than on the 16th May. For the Ebble, there is a saddle at approximately one hour after sunset where there is a sudden drop in the rate of decline. The main feature of the $\Delta DO$ plots, however, is the difference in timing of the point at which $\Delta DO = 0$, where for the Ebble and the

Nadder it occurs between 1 and 3 hours after sunset, for the Avon between 6 and 8 hours after sunset, but for the Wylye on the 9th it occurs early at 3 hours after sunset and on the 16th it occurs late at 7.5 hours after sunset.

   Identification of the point at which there is zero change in DO is not as straightforward as at first it seems. For the Ebble, for example, the change in DO for any time step is close to, but never equal to, zero. Identification could be achieved by fitting a line to the points in Figure 4 and noting where the line crosses $\Delta DO = 0$, but this

presupposes a particular model structure which may be invalid. There are two other approaches, both of which have limitations and both of which were implemented as a mutual check. One method (Method 1) is to locate the point during the night at which DO is at a minimum. One limitation is that there may be multiple local minima because of short term DO fluctuations, any of which could be the 'true' global minimum for that night. The main limitation, however, is that DO may decrease throughout the night such that minimum DO occurs at the end of the night and

$\Delta DO$ itself is never equal to zero. Therefore, as a safeguard, in the implementation of Method 1, if the minimum DO was found to occur within 20 minutes of sunrise, that outcome was discarded. The third approach (Method 2) is to compare each pair of contiguous data points in the smoothed DO time series and identify those points where $\Delta DO$ changes sign. These are shown as black triangles in Figures 4 and 5, which gives a range of $DOD_{zero\ \Delta DO}$ values. For example, for the Wylye for May 16th/17th there are 11 data points where $\Delta DO$ changes sign, with associated

values of the DO deficit ranging between 3 and 3.18 mg DO per litre with a median value of 3.06 occurring at 3 hours and 9 minutes after sunset. For the Ebble, corresponding numbers are 1.57 to 1.73 with a median of 1.7 mg DO per litre occurring at 2 hours and 46 minutes after sunset. The median value of those points can then be taken as the single value of the DO deficit where $\Delta DO = 0$. The drawback of this approach is that there may be anomalous data points (for example the Nadder in Figure 5), which might yield erroneous $DOD_{zero\ \Delta DO}$ values. For

the two nights, the sets of $DOD_{zero\ \Delta DO}$ values are shown as boxplots (Figure 6). Also shown (black triangles) are





the corresponding values of the regression quotient calculated from the night-time regression method. For the Ebble and the Nadder (panel (A)), the regression quotient underestimates the range of $\text{DOD}_{zero\ \Delta DO}$ values. For the Avon (panel (B)), the regression quotient overestimates the median $\text{DOD}_{zero\ \Delta DO}$ value. For the Wylye (panel (B)), the regression quotient overestimates on the 9th/10th and underestimates on the 16th/17th. Thus, on the night when

the DO minimum comes early after sunset and the Wylye behaves more like the Ebble and the Nadder in terms of timings of DO dynamics, the regression quotient underestimates the median $\text{DOD}_{zero\ \Delta DO}$ value.

For data covering the entire study period (August 2014 to August 2015), the distribution of the ratio of median $\text{DOD}_{zero\ \Delta DO}$ values to the regression quotient is shown for each river in Figure 7. Where the ratio is greater than 1, the median $\text{DOD}_{zero\ \Delta DO}$ exceeds the regression quotient. For the Ebble and Nadder this is the case for about

three quarters of the nights (75% and 73% for Ebble and Nadder respectively) whereas the distributions are more symmetrical for the Wylye and Avon with corresponding proportions of 60% and 44% respectively. This data is shown as a set of time series in Figure 8, together with a time series of the comparison with the regression quotient. For the Ebble, median $\text{DOD}_{zero\ \Delta DO}$ values range between 1 and 2.5 mg DO per litre with two peaks, one in Oct/Nov 2014 and a second in summer 2015. A trough occurs in winter, before rising to values in May similar to those in the

previous September. For the Wylye, values range between 2 and approximately 5 mg DO per litre; data for June 2015 onward are more volatile and consequently less clear with regard to any evident pattern. The seasonal pattern differs in that there is no Nov 2014 peak, with an earlier autumnal peak occurring in Sep 2014. Values for the Avon range between between 2 and 4 mg DO per litre with peaks in Oct/Nov 2014 and a second in Jun/Jul 2015. From mid-March to mid-April and again in late May/ early June, the median $\text{DOD}_{zero\ \Delta DO}$ for the Avon were persistently

low with values of about 1 mg DO per litre or less. These points were considered anomalous and were discarded from the analysis. For the Nadder, the median $\text{DOD}_{zero\ \Delta DO}$ value rises steadily from a value of 1.5 mg DO per litre at the beginning of March 2015 to approximately 2.3 mg DO per litre in late June 2015. The Nadder differs from the other three sites in that there is only one peak occurring between May and September 2015, although the caveat is that data for the first part of the time series is missing. None of the sites shows a marked difference in behaviour

according to whether Method 1 or Method 2 is used.

Also shown (Figure 8) is a comparison between median $\text{DOD}_{zero\ \Delta DO}$ value ($\frac{R}{k}$) and the regression quotient ($\frac{R_{HK}}{k_{HK}}$), expressed as the ratio of the former to the latter. For the Wylye and the Avon, this ratio is very close to 1 over most of the year. For the Ebble and the Nadder, however, the median $\text{DOD}_{zero\ \Delta DO}$ values almost always exceed the regression quotient. The notable exception is in Oct/Nov 2014 for the Ebble where this pattern is reversed with the

regression quotient tending to exceed median $\text{DOD}_{zero\ \Delta DO}$ values. Whether the regression quotient overestimates or underestimates median $\text{DOD}_{zero\ \Delta DO}$ value depends partly on when median $\text{DOD}_{zero\ \Delta DO}$ ($\frac{R}{k}$) occurs as shown for the Wylye in Figure 9; if the change in sign of $\Delta$ DO occurs relatively quickly after sunset (between 2 and 6 hours after), then the regression quotient underestimates median $\text{DOD}_{zero\ \Delta DO}$ and as time after sunset increases, the regression quotient has a tendency to overestimate $\text{DOD}_{zero\ \Delta DO}$ values. As time after sunset further increases, the

regression quotient again underestimates median $\text{DOD}_{zero\ \Delta DO}$. To demonstrate that this is not simply a seasonal





effect, this pattern is shown for the entire study period (panel A) and also for the two month period up to May 20th 2015 (panel B).

Figure 10 shows a time series for each river relating to the length of time after sunset at which median $DOD_{zero\ \Delta DO}$ occurs. For September 2014 to February 2015, this interval is notably variable for all rivers, ranging between 2 and

10 hours. For May to July 2015, the Ebble and Nadder show a clear pattern of a reduction in time to $DOD_{zero\ \Delta DO}$. For the Nadder, this remains relatively constant at between 2 and 3 hours. For the Ebble, DO reaches its minimum point most quickly in May at approximately 3 hours after sunset, but then rises steadily through approximately 4 hours in June, 5 hours in July and 6 hours after sunset in August. For the Wylye, $DOD_{zero\ \Delta DO}$ in May and June 2015 occurs typically at just under 5 hours after sunset. Despite the fact that at other times of the year, the time

interval is more variable, nevertheless the annual pattern as indicated by the trend line, shows a clear periodicity with a maximum of approximately 10 hours in winter (Nov to Jan) for all rivers with river-specific patterns in spring and summer.

The regression quotient up to this point was computed using all data points for any given night. An alternative would be to calculate the regression quotient using only a subset of night-time points. One possibility would be to do

so using only those data points clustered around the time after sunset at which $\Delta DO = 0$. The effect of this is shown for the Nadder in Figure 11, which compares the regression quotient for each night in the year, calculated using all data points for each night, with that obtained using only those data points that are recorded within 15 minutes either side of the time where $\Delta DO = 0$. By restricting the night-time regression calculation to those points, the bias is seen to be removed. This does not necessarily mean that the associated estimates of R and k are better, but

it might mean that comparisons between nights are more consistent, although this possibility was not investigated further.

**Discussion**

For four sites on four separate rivers, two Chalk (Wylye and Ebble) and two Greensand (Avon and Nadder), the night-time dissolved oxygen deficit at points of zero DO change was identified and used as a proxy for the ratio

of community respiration to the aeration rate constant. This ratio was compared to a theoretical equivalent, the regression quotient, computed using the night-time regression method (Hornberger and Kelly, 1975). The time after sunset at which DO change is zero was also noted and it was found that for rivers where that happens early during the course of the night, the regression quotient underestimates the $DOD_{zero\ \Delta DO}$. This was summarised for the river Wylye, which showed for the year as a whole that when $DOD_{zero\ \Delta DO}$ occurs early in the night the regression

quotient underestimates $DOD_{zero\ \Delta DO}$ and when it occurs late, the regression quotient overestimates. It was also shown that DO magnitudes for the two Chalk rivers are similar but timings for the Ebble were very close to those of the Nadder, including the time at which DO peaks in the daytime. When median $DOD_{zero\ \Delta DO}$ values were





compared to the regression quotient for the year as a whole, the regression quotients for the Ebble and Nadder persistently underestimated median $\text{DOD}_{zero\ \Delta DO}$ values.

Different streams behave in different ways. The standard, single stage respiration diel oxygen model assumes constant R and constant k (notwithstanding temperature effects) over a 24 hour period. If this assumption is true,

then median $\text{DOD}_{zero\ \Delta DO}$ should be equal to the regression quotient. For the Wylye and Avon, the $\text{DOD}_{zero\ \Delta DO}$ values are close in value to the regression quotient, but for Ebble and Nadder they show more divergence. However, the standard diel oxygen model is typically applied uniformly across streams. This means that when quantitative inferences are made through use of the model, they may be more appropriate for one stream than the other. If differences are then found between rivers (with regard to primary production and respiration for example), it will

be presumed that the differences arise because of differences in the substantive behaviour of the river, that the differences are real. In fact, it could be that primary production is the same in both cases, but differences arise because of inappropriate use of the model. Therefore, it is important to identify the correct model for each river system and indiscriminate application of the single diel oxygen model can result in misleading inferences when comparing different sites.

In fact, recent research (Schindler *et al.*, 2017) indicates for some streams the existence of separate autotrophic and heterotrophic respiration components with two sources of organic matter, one connected to recent photosynthetic activity and a second, ambient, time-invariant source. They suggest that increases in night-time oxygen concentrations, as is the case for both the Ebble and the Nadder in May (Figures 4 and 5), might be indicative of two-stage ecosystem metabolism. For the data presented here, this is a feasible explanation. The fact that the Ebble and

Wylye exhibit similar DO ranges in May, yet the daytime Ebble DO peak typically occurs earlier suggests that for the Ebble, as photosynthesis progresses, some products of that process are aerobically consumed.There is, however, an important caveat. Of the four rivers, the Wylye records the highest DO concentrations so that there is a *prima facie* case that the Wylye is the most productive. Although there were nights during which the Wylye showed increases in nighttime DO, yet it still consistently recorded highest daytime DO values in May 2015. Assuming that this is

because the Wylye has highest primary production, that would mean that nighttime rises in DO maybe sufficient, but not necessary, as indicators of productive aquatic systems.

Behaviour of DO dynamics were also examined with regard to hours after sunset at which $\Delta\text{DO}$ is zero. Typically, a DO time series will be presented with time marked as civil time in a particular time zone, but framing time in terms of the behaviour of the sun both makes inter-site comparisons more transparent and also is more pertinent to

30 the response of the aquatic plant community. It also means that, by identifying as an annual time series, the time after sunset at which $\Delta\text{DO}$ is zero, anomalous behaviour can also be identified and used either as a filter to remove spurious data or as a flag to search for particular events, for example floods or periods of unusually low flow.

The regression quotient was calculated for the night as a whole and also by restricting the data points included to those $\pm\,15$ minutes either side of the point at which change DO is zero. This was found to remove the bias. This does

not mean that calculations of R and k using only those points will give better estimates of R and k, since if there is





two stage metabolism, then such an approach would be disregarding the photosynthetic-dependent R, although it might mean that intra-stream comparisons over a series of nights are more consistent.

**Conclusion**

This paper began with a comment on the proliferation of automatic logging devices which vastly increases the potential for analysis of river oxygen and therefore river carbon dynamics. Oxygen dynamics are often analysed using models that make simplifying assumptions about the underlying processes, specifically about the constant values of both community aerobic respiration and the reaeration rate constant over the course of a single day. However, there is a debate about the extent to which respiration in particular can be represented by a single daily value. Through analysis of the dissolved oxygen deficit at points of zero DO change for four sites on four rivers, it was shown here that the assumption of constant values for either respiration or the aeration rate constant was violated perennially for two of those sites. It was suggested that this is likely to be because of two stage rather than one stage respiration, although it should be noted that possible variability in the aeration rate constant over the course of a single day was not considered. In any case, this means that the use of single station, single stage respiration diel oxygen models might not be optimal in such cases. This is not to decry the use of such models, as the purpose of a model is to abstract from reality. However, if analysis of DO time series were to become routine with results impacting environmental policy decisions, then it is important to understand when these models are failing rather than presume that they are fit for purpose.

*Data availability* Data are stored with the Natural Environment Research Council and will be made publicly available after September 2019.

*Acknowledgements* This work was funded by the Natural Environment Research Council (NERC) Macronutrient Cycles thematic programme (grant numbers NE/J012106/1 and NE/J01219X/1)).

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



**Table 1.** Site location and catchment characteristics

| River | Major geology | Latitude | Longitude | Catchment size (km$^2$) | BFI | Mean flow (m$^{-3}$s$^{-1}$) (July 2014 to June 2015) |
|-------|---------------|----------|-----------|-------------------------|-----|-------------------------------------------------------|
| Ebble | Chalk | 51.028 | -1.924 | 58.9 | 0.906 | 0.60 |
| Wylye | Chalk | 51.143 | -2.203 | 53.5 | 0.901 | NA |
| Nadder | Greensand | 51.045 | -2.110 | 34.6 | 0.781 | 0.40 |
| Avon | Greensand | 51.319 | -1.862 | 59.2 | 0.744 | 0.45 |

Sources: Heppell *et al.*, 2017 and for flow data Heppell and Binley, 2016.







**Figure 1.** DO time series for May 5th to May 20th 2015. Solid grey areas are the nights of the 9th/10th and 16th/17th May.





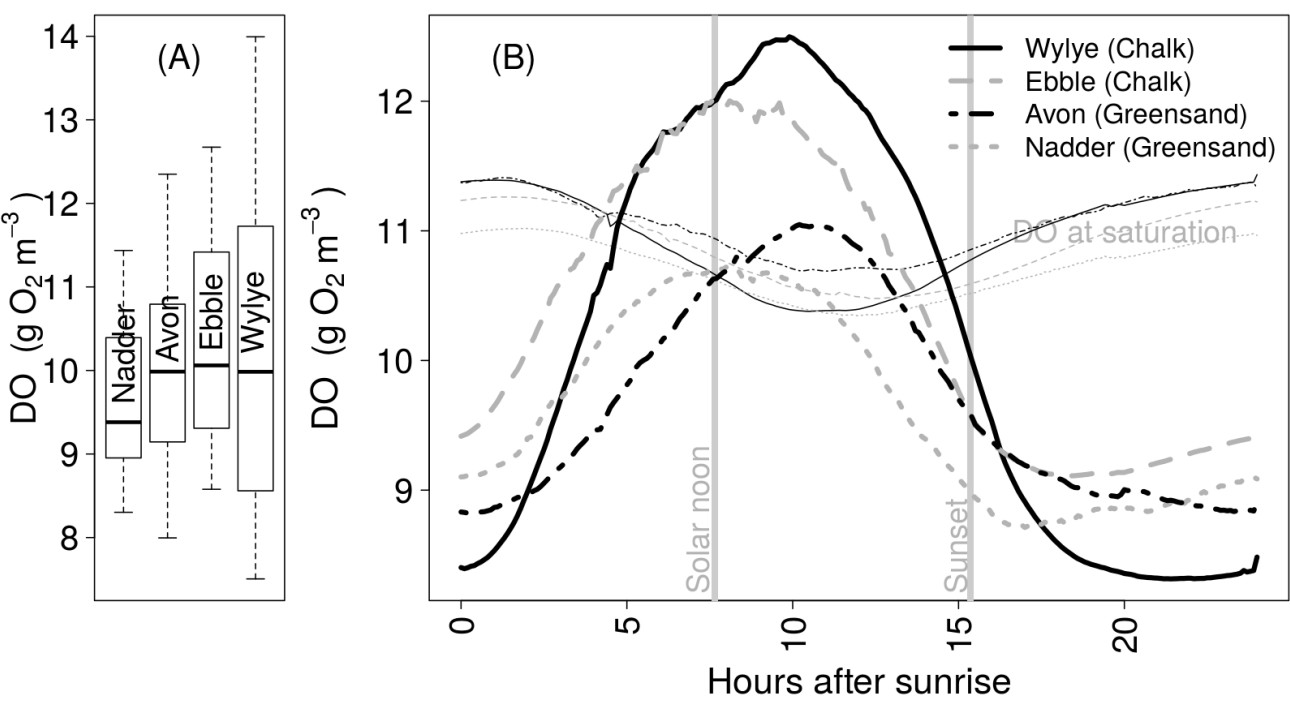

**Figure 2.** Distributions of DO values (A) and mean DO by hours after sunrise (B) for May 5th to May 20th 2015.

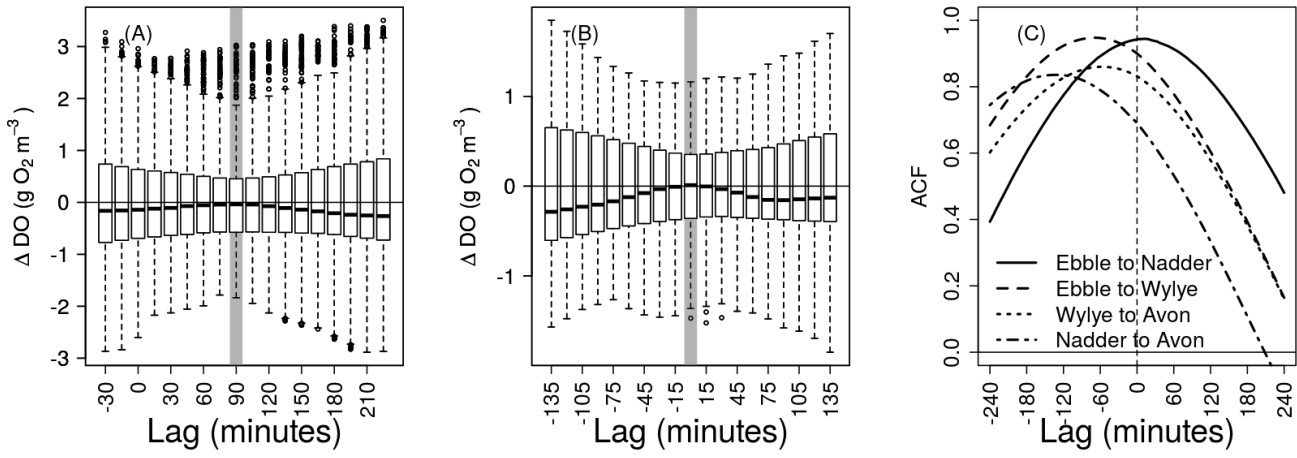

**Figure 3.** Analysis of lagged differences in DO between detrended DO time series. (A) Ebble and Wylye, (B) Ebble and Nadder (C) cross-correlations for four rivers for May 5th to May 20th 2015.





**Figure 4.** Time series for DO and ΔDO for the night of 9th to 10th May. ΔDO is at one minute intervals. Bold triangles mark those points where there was a change in sign of ΔDO.





**Figure 5.** Time series for DO and ΔDO for the night of 16th to 17th May. ΔDO is at one minute intervals. Bold triangles mark those points where there was a change in sign of ΔDO.





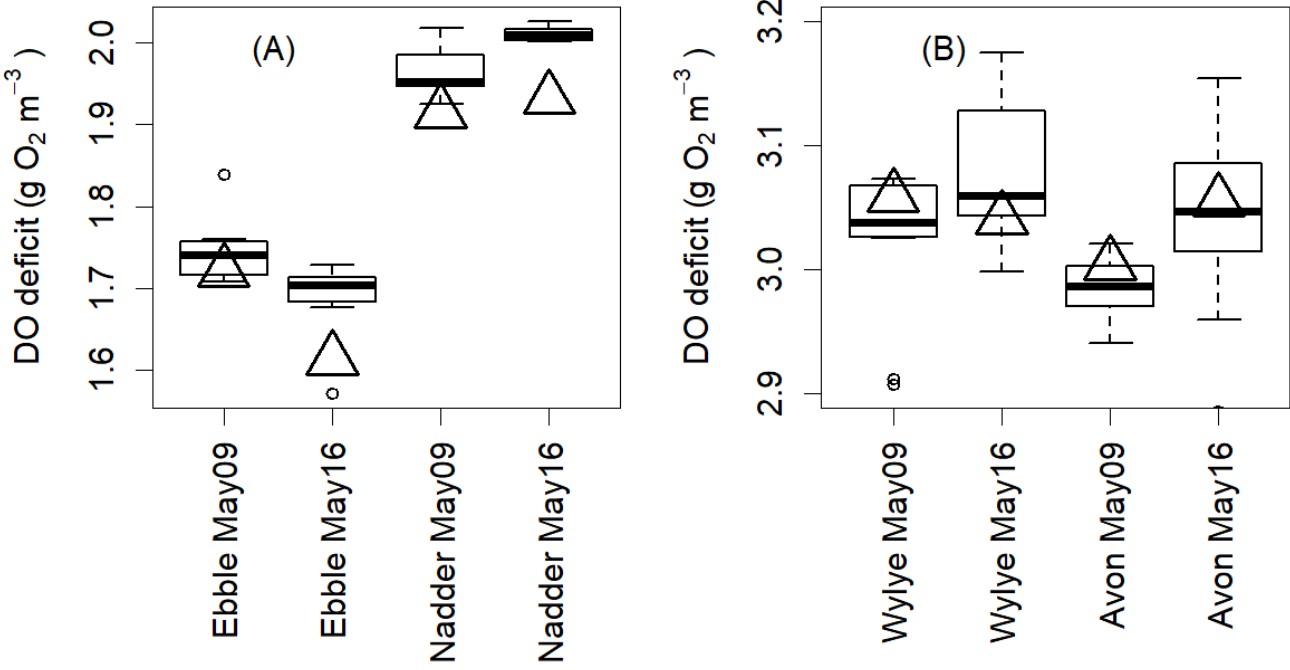

**Figure 6.** Boxplot time series of DO deficits at points of steady state DO for the nights of May 9th/10th and May 16th/17th 2015. Values for the regression quotient are shown as triangles.





**Figure 7.** Distributions of the ratio of DO deficit at points of zero DO change to the regression quotient.

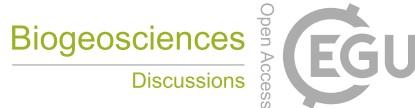



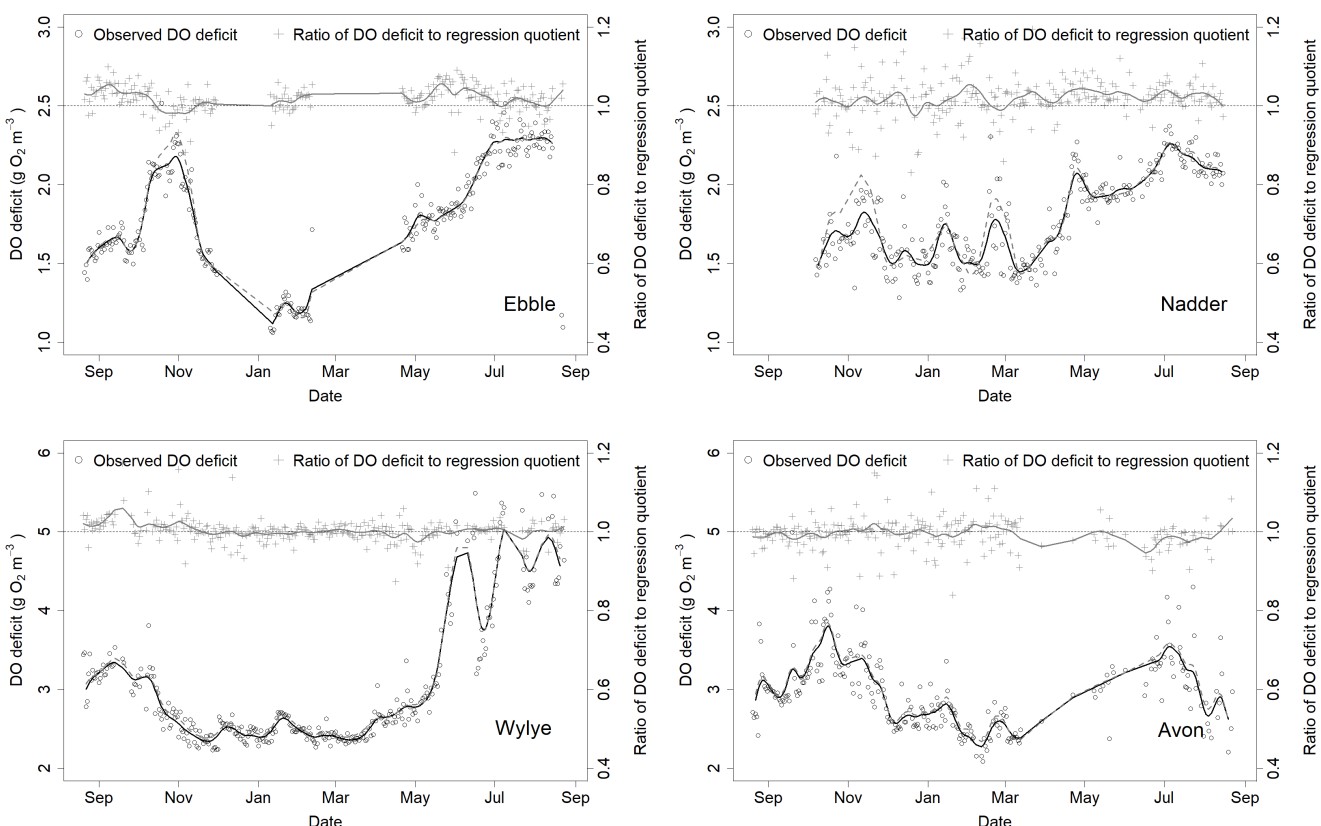

**Figure 8.** Time series of the DO deficit at points of zero DO change (black circles) and comparison with corresponding ratio derived from night-time regression (grey crosses). Trend lines are shown for both time series. For grey dashed line (Nadder), see text.





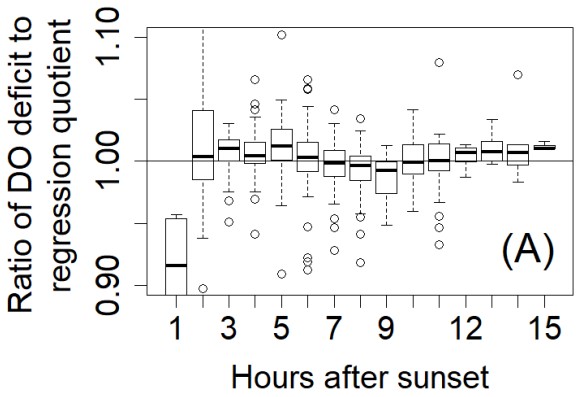
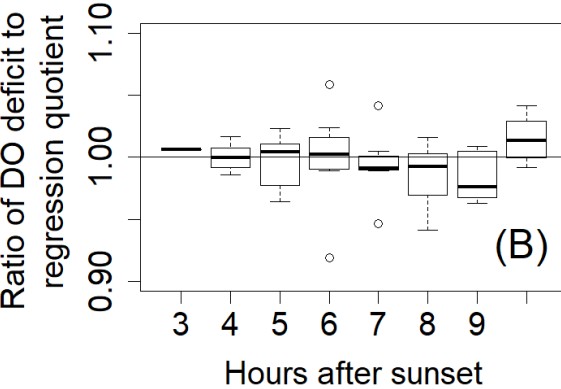

**Figure 9.** Relationship between time after sunset until point of zero DO change and the ratio $\mathrm{DOD}_{zero\ \Delta DO}$ : (*regression quotient*) for the river Wylye for the entire study period (A) and two month period up to 20th May 2015 (B). Hours after sunset are rounded to the nearest whole hour.





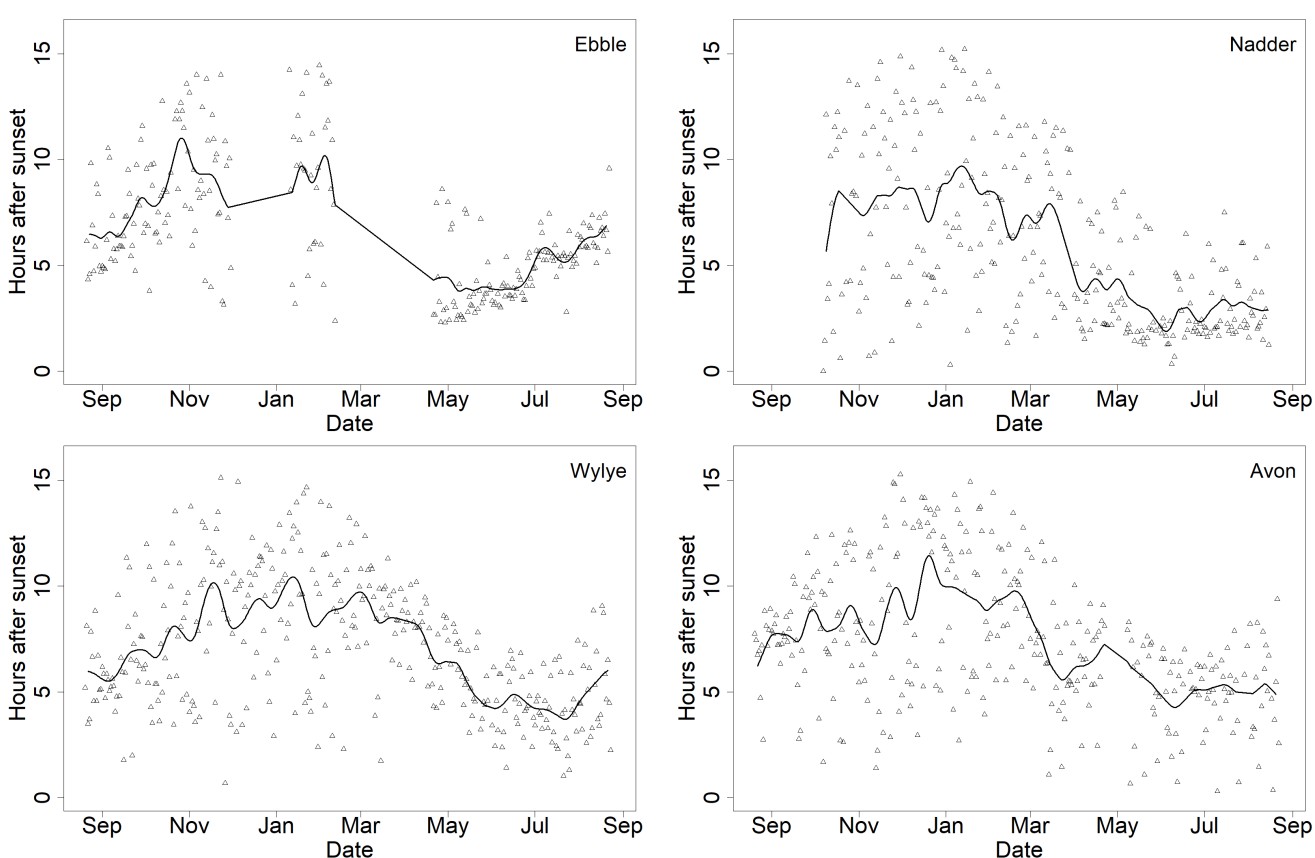

**Figure 10.** Time after sunset at which ΔDO is zero.





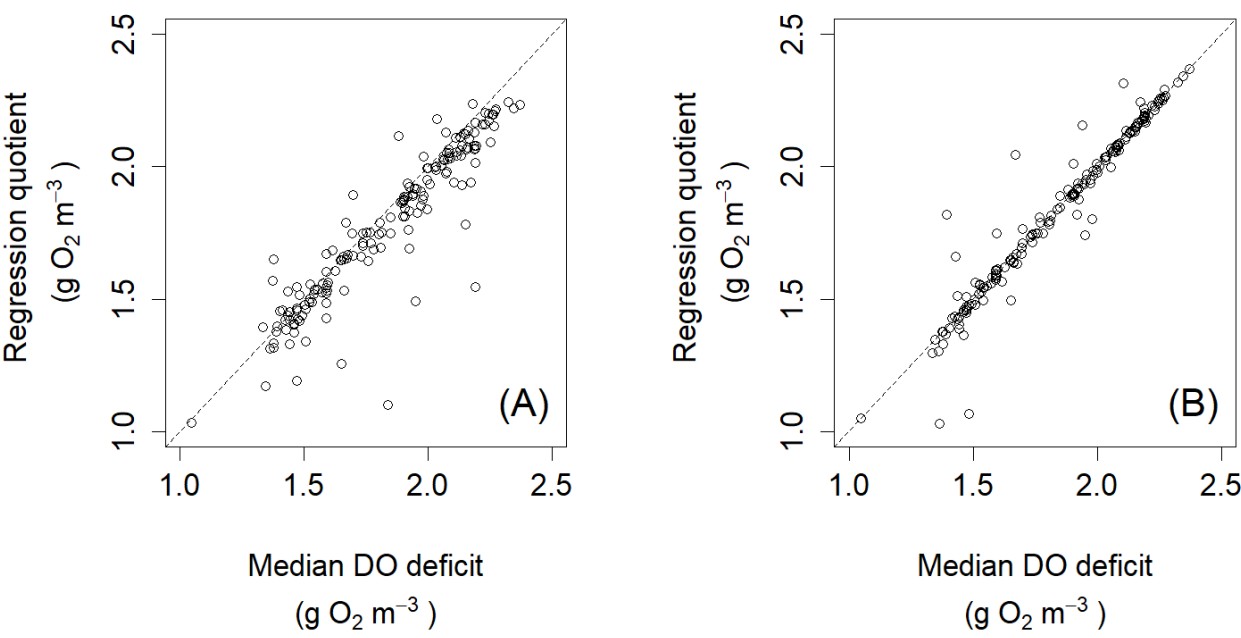

**Figure 11.** Scatter plots for regression quotient against median DO deficit at points of zero DO change for river Nadder.