# Peer review of "Oxygen dynamics and evaluation of the single station diel oxygen model across contrasting geologies"

_Biogeosciences, 2019_

## Referee Comment (RC1) · Anonymous Referee #1 · 12 Jun 2019

This manuscript uses a unique approach to evaluate whether the assumption of a constant rate of ecosystem respiration is valid over a daily cycle as assumed in most aquatic ecosystem metabolism models. He evaluates whether the point at which the rate of change in oxygen concentration for a given day is equal to zero (i.e., dO2/dt = 0) provides information about the ratio of ER/k within and across stream types. He then argues that because this technique does not agree with results from the nighttime regression approach of Hornberger and Kelly (1975) that the assumption of a constant daily rate of ER is invalid.

However, beyond that, it is not clear to me how this approach provides an estimate of

diel change in R, as stated in the abstract and discussion? We know that ER changes over the course of a day in response to temperature (Holtgrieve et al. 2010) and carbon substrates (Schindler et al. 2017, Sadro et al. 2014) but you generate a single ratio using this approach, not explicit rates of ER.

In addition, I'm not sure what the ratio (ER/k) really describes – how does this get you additional information that you don't get by fitting a metabolism model, since the ratio of R/k doesn't give you any information on their relative magnitudes. And, wouldn't you still face the issue of equifinality (many values of ER and k that could produce a given ratio)?

Further, you discuss the importance of correcting ER and k for temperature, but then don't consider that in your estimation of their ratio – wouldn't the diel variation in temperature have a lot to do with when the point of dO/dt = 0 occurs as well? In addition, the temperature correction is different for the two, so the degree of daily temperature fluctuation could impact the resulting ratio.

Zero change in DO has an equal element of uncertainty to it (when does DO/dt = 0?) as does fitting a nighttime regression (i.e., where does night begin?) so I'm not sure what you gain through using this approach? In addition, using the nighttime regression technique is no longer the most common way of estimating reaeration rates because of some of the shortcomings you mention and cite.

In summary, I have concerns about the significance of these findings given the degree to which the field has moved on in terms of approaches to metabolism models (e.g., Appling et al. 2018, Song et al. 2016) and the ability to estimate and constrain k (Appling et al. 2018, Raymond et al. 2012). In addition, the results as presented spend a significant amount of time discussing the degree of coherence in diel oxygen patterns (e.g., timing of max O2), rather than comparing potential daily fluctuations or cross-system differences in important metabolic parameters. I recognize that assessing differences in the magnitude and timing of daily fluctuations has some meaning in

terms of understanding the magnitude of processes, but this discussion would be more meaningful had those processes also been quantified.

Specific comments: Page 2, Lines 3-4: "Primary production can be quantified by partitioning a single DO time series into its component fluxes, namely photosynthesis, ecosystem respiration and aeration" – perhaps you mean to say "ecosystem metabolism" Page 3, lines 3-5: What is the "pertinent process"? Page 3, lines 19-20: "primarily groundwater fed". . . couldn't the point of zero change reflect the O2 concentration of groundwater input as well? Page 4, lines 27-28 – wouldn't it be "normalized" not "detrended"?

―――――――――――――――――――――――

---

## Author Comment (AC1) · 22 Jun 2019

The rationale of the paper is to start with data and avoid making assumptions about model structure. It is possible to observe (i.e. as data) when dO/dt is zero as shown in figure 4, and although it sometimes might not identify the correct point, as in Nadder fig. 5, it is a robust and reproducible way of identifying this point if you have a sufficiently high frequency of measurement. From that it is then possible to calculate the saturation deficit. The important point is that I have not assumed any model structure to do this, so this is a data-driven approach.

To the best of my knowledge, there are three model structures which are used to quan-

tify stream metabolism:

(1) dO/dt = P - R + k(DO_sat - DO_actual)

(2) As (1), but with temperature correction for P, R and k. (e.g. Correa-Gonzalez et al., 2104; Song et al., 2016, Appling et al., 2018; Richmond et al., 2016)

(3) As (2), but R is split into an ambient component and a component linked to photo-synthesis (as in Schindler et al., 2017)

Most researchers use (2).

But Schindler et al.(2017) state: "The two-stage model fit oxygen data considerably better than a single-stage model in nine of 13 stream x date combinations we considered (Table 2)." (p.13)

Appling et al. (2018) justification for use of a simple model is: "More complex equations describing nonlinear relationships [...] may be useful in modeling some rivers [..however..] here we use simple models that are more resistant to overfitting and are likely sufficient for many streams" but what proportion of streams are many streams in "...likely sufficient for many streams"?

Song et al., 2016 state: "Changes in DO concentration can be generally described by [Equation 2]." But how do they know this is the case?

So the question is, how do you distinguish between choice of model structures?

That is the purpose of comparing the dissolved oxygen deficit at zero DO change with the corresponding ratio calculated using Hornberger-Kelly, because it is a metric for how far the assumptions used in a simple model deviate from a quantity which can be measured. If the deviation is large, then you should consider rejecting your choice of model structure.

It's an obvious thing to do, but it has not been done.

---

## Author Comment (AC2) · 22 Jun 2019

Thank you for reading the paper and comments.

I have separately written a general comment that addresses objectives, as I think perhaps they weren't clear. I should say that there was never any intention to quantify R or k. The idea was to examine what can be measured rather than what can be modelled.

Specific points:

"...it is not clear to me how this approach provides an estimate of diel change in R, as stated in the abstract and discussion?" I don't state this in the abstract or the discussion. I do not make any attempt to quantify R or claim that I am doing so. The whole approach is about evaluating model structures, not quantifying parameters.

"In addition, I'm not sure what the ratio (ER/k) really describes – how does this get you additional information that you don't get by fitting a metabolism model". This is the central issue. Which model do you fit and if you choose the simple model (single stage ER) how do you know it's good enough?

"...so the degree of daily temperature fluctuation could impact the resulting ratio." Yes, this is true that I do not explicitly consider temperature (although I do state that I'm not considering temperature). In other words, you are saying that I am evaluating model structure 1 (in the separate author comment) rather than model structure 2 (which is what most people use). In response to your comment, I have in the past week done additional analysis which compares the temperature regimes for the two nights (May 9th and May 16th) for all four rivers which demonstrates (to me, at least) that temperature is not the dominant control. The figure is attached. There is no evidence that the Ebble and Nadder have one temperature regime and the Wylye and Avon have another (which would be the case if temperature was the explanation for the differences in behaviour shown in Figure 7 and figures 4 and 5). In fact, the temperature regimes for the Nadder and Avon are more similar to one another than those for the Nadder and Ebble, even though the Nadder and Ebble are the rivers with early DO nighttime minima, so it seems to me at least that temperature is not the explanation.

"Zero change in DO has an equal element of uncertainty to it (when does DO/dt = 0?) as does fitting a nighttime regression (i.e., where does night begin?)". I disagree with this statement. If you have sufficiently high resolutiion of measurement (1 minute in this case), the uncertainty associated with when dO/dt = 0 depends only on the accuracy and precision of the instrument, and the occurrence of spurious data points as with the Nadder in Figure 5. Otherwise, it's robust and reproducible. With regard to uncertainty of nighttime regression, yes, it does depend on what portion of the night you use, but I discussed this is in Figure 11. It seems to me that the important point is that the relationship between the two differs systematically between rivers. So, for example, even though the Wylye and Ebble are both Chalk and groundwater-dominated to the same extent in that they both have BFI of 0.9 (Table 1), nevertheless the relationship between DO saturation at zero DO change and the corresponding ratio calculated using Hornberger-Kelly is completely different (Figure 7). A corresponding argument applies to the Nadder and Avon.

"...and the ability to estimate and constrain k (Appling et al. 2018)." Yes, but their justification for the use of simple models "that are more resistant to overfitting and are likely sufficient for many streams" is axiomatic and based on little evidence, as far as I can see. If the model structure is inappropriate, there is not much point in constraining k.

I am not saying that simple models are never appropriate or that respiration can never be approximated by a single stage process, but the question is when are they appropriate?

What I am trying to do is provide an easily measured metric that can be used as an additional aid to give some indication as to the appropriateness of model structure. It's not a silver bullet, it's just an additional way of characterising the system.

Response to specific comments: Yes, ecosystem metabolism. Pertinent processes are those in the equation. In other words, assuming that the equation captures the oxygen behaviour and no significant processes have been omitted. Groundwater. This is addressed above. Despite the fact that the rivers have very similar groundwater regimes, nevertheless they have differing oxygen regimes as described, so that the difference in oxygen behaviour cannot be attributed to differences in groundwater regime. Yes, normalised.
* * *
[Figure]

**Fig. 1.**

---

## Author Comment (AC3) · 22 Jun 2019

The other thing which to some extent has been overlooked is that the paper is not just about the appropriateness of metabolism models. Streams are sometimes referred to by their geology (in the UK at least). It might be presumed that because one stream is similar to another in terms of geology and groundwater regime, then they would be similar in terms of ecological functioning. But in this case, in terms of oxygen dynamics, one Chalk stream has behaviour similar to a Greensand stream, rather than to another Chalk stream. For me, this was unexpected.

---

## Referee Comment (RC2) · Anonymous Referee #2 · 4 Jul 2019

I'm sorry this review is delayed but the delay has resulted in clarification of a number of my questions which were picked up by the other referee. In particular, I now have a much better understanding of the purpose of the paper. This was not clear on the initial reading.

I would therefore strongly recommend that the abstract be revised to better reflect the purpose of the paper. i.e. the point stressing that this paper is " about evaluating model structures, not quantifying parameters", needs to be right up front.

It is also worth reflecting on George Box's pithy aphorism that all models are wrong but some are useful. This is highly relevant to this manuscript as it begs the question

'useful for what?' There are a number of generalizations and simplifications implicit in solving the 'simple' metabolism model of Odum into three components viz. (ecosystem) respiration, primary production and reaeration. The author points out the likely effects of autochthonous vs allochthonous carbon on respiration rates and how this is likely to be time dependent on a daily time frame. I totally agree with this. The key point for me though is 'does this matter?' What question is driving the use of stream metabolism in the first place? If it's mechanistic understanding, then nuances matter very much. If it's about aggregating organic carbon loads across time and space, contrasting watersheds with different land use for example, then it doesn't matter anywhere near as much, if at all.

As a general point, I find discussion centered on changes in DO based on temperature often inadequate as the obvious effect of temperature effect on DO solubility is neglected. An increase in night time DO is expected if water temperature falls. Framing the discussion in terms of %DO saturation is much more useful to examine the interplay of R and k but in this instance makes it more convoluted to then talk about change in DO = 0. Of course if temperature doesn't change (nor atmospheric pressure or salinity to be pedantic) then change in DO = 0 would correspond to a change in %DO saturation of 0.

Minor point: Page 2, Line 4. Suggest changing 'components' as this word was used in the previous sentence to refer to P, R & reaeration.

Because this is novel, I would like to see a little more explanation about HOW R/k can be used to interrogate the validity of the model used for fitting diel O2 curves. This may be immediately obvious to those who routinely inverse model metabolism, but to many readers this won't be clear at all.

Page 2, Line 20. Equifinality hinders resolving ER and k when %DO saturation is very close to 100% or when there is very little change in %DO over the day, it is not a universal problem.

Page 3, Line 26. It has already been stated that precision and accuracy of the DO sonde data is of fundamental importance in reliably identifying points where the change in DO is zero. Yet there is no mention at all of how accuracy of the deployed sondes was verified. Probe drift would be a major confounding factor in this analysis.

Page 5, Line 13. The cause of this sudden change in the rate of decline?

Is there any significant time-of-day dependent topographic shading of any of these streams over the study reach integrated by the sonde? If so, this may then affect time of peak DO.

Reaeration will not only depend on temperature (in a well-known relationship) but also on wind (there is a lot of lakes' literature on this topic) and on discharge. A change in discharge will almost certainly change k and this relationship will be idiosyncratic for each site depending on stream channel shape, wetted area, roughness etc. Are these additional factors responsible for some of the variation observed in this data set?

Bearing in mind the already posted review and the author's responses, I still believe this is an interesting paper that definitely warrants publication. However, to be more useful to the general reader and in particular those undertaking studies where stream metabolism is being measured, I strongly recommend the author provides an additional paragraph or two which guides the user through checking the inherent model assumptions when modelling their data. This can be in a series of steps checking whether the assumption of an invariant R (temperature effects notwithstanding) has a significant effect on overall metabolic parameter estimates.

---

## Author Comment (AC4) · 24 Jul 2019

Thank you for reading paper and helpful comments.

"I would therefore strongly recommend that the abstract be revised to better reflect the purpose of the paper. i.e. the point stressing that this paper is " about evaluating model structures, not quantifying parameters", needs to be right up front."

Ok.

"Minor point: Page 2, Line 4. Suggest changing 'components' as this word was used in the previous sentence to refer to P, R & reaeration. "

Agreed.

"I would like to see a little more explanation about HOW R/k can be used to interrogate the validity of the model used for fitting diel O2 curves."

Assume both R and k are constant. And given (at nighttime):

d(DO)/dt = -R + k(DOsat - DO) [Equation 1]

Then:

(1) A plot of d(DO) against (DOsat - DO) will give a straight line with slope k and constant term R. This is used to calculate a ratio R/k (ratio 1)

and:

(2) When d(DO)/dt is zero, (DOsat - DO) is measured. This gives a different method of calculating the same quantity, R/k (ratio 2).

If ratio 1 equals ratio 2, then Equation 1 adequately describes the nighttime DO dynamics. If, however, they are not equal, then Equation 1 does not adequately describe the processes.

For the 16th May, for example, for the Ebble ratio 1 is 1.6 and ratio 2 is 1.7. But for the Avon, they are equal (3.05).

If you then look at the simulations, the figures below (Figure 1 and Figure 2) show optimised models for the night of the 16th of May for the Ebble and the Avon. Grey circles are observations. Grey line is simulation not accounting for temperature. Black line is simulation accounting for temperature (i.e. of the form, R = R20 ˆ(T-20), where R20 is respiration at 20 degrees C, T is temperature in Celsius). The right hand panel shows the residual plots (observed DO minus simulated DO). The fit for both (Ebble and Avon) is good, but the residuals for the Ebble show that the data depart from the model in a systematic way.

This does not demonstrate that R is not constant, but demonstrates that the assumptions are not upheld and it is a measure of the extent to which the model deviates from the data.

"Page 2, Line 20. Equifinality hinders resolving ER and k when %DO saturation is very close to 100% or when there is very little change in %DO over the day, it is not a universal problem."

If changed to 'can be hindered', is that acceptable?

"Page 3, Line 26. It has already been stated that precision and accuracy of the DO sonde data is of fundamental importance in reliably identifying points where the change in DO is zero. Yet there is no mention at all of how accuracy of the deployed sondes was verified. Probe drift would be a major confounding factor in this analysis."

It's true that drift could be a confounding factor, so I've done some additional analysis (Figure 3). I have made the following assumptions. The probe has drifted over some unspecified period by 0.5 mg DO per litre. Drift over the course of any single night is negligible. Temperature has not drifted (although I could test for this also).

The effect of this is to reduce the DO deficit at zero DO change from 1.7 to 1.2 mg DO per litre and the corresponding Hornberger-Kelly ratio from 1.61 to 1.11, so even if there were drift it wouldn't affect the conclusion. The test (comparing Hornberger-Kelly ratio with DO deficit) most likely examines the shape of the DO curve; it's not about magnitudes (I think).

"Page 5, Line 13. The cause of this sudden change in the rate of decline?"

I don't know. I could speculate that it is because labile organics have been consumed, but that would be too convenient for the overall argument (although that is a possible explanation). It is a long time series (each river is about half million records), so it's hard to explain local features.

"A change in discharge will almost certainly change k and this relationship will be idiosyncratic for each site depending on stream channel shape, wetted area, roughness etc. Are these additional factors responsible for some of the variation observed in this data set?"

Could it be k which is not constant? Of course, it's possible that k is different across sites, but could k change during the course of one night according to the same pattern for several nights in a row? I have attached a plot for discharge (Figure 4). There is no difference in the discharge that would account for differences between Ebble and Avon. It could be that windspeed is changing every night in the same way and therefore k is changing, but changes in the windspeed would be similar across all sites. Therefore, changes in windspeed could only account for the behaviour if the Wylye and Avon were sheltered, and buffered from the effects of changes in windspeed. Yes, this is possible and cannot be ruled out. On the other hand, windspeeds tend to drop during the night, so that, if for the Ebble and Nadder, the explanation for variable k were falling windspeed, then you would expect DO to stagnate as the night progresses, but the reverse is the case.

"Is there any significant time-of-day dependent topographic shading of any of these streams over the study reach integrated by the sonde? If so, this may then affect time of peak DO."

Yes, this is true. This cannot be ruled out. Also, if time to peak is shorter duration, then time to minimum (after sunset) is likely (although not inevitably) to be shorter duration. But early time to peak (and early time to minimum) for both Nadder and Ebble together with the fact that it is those two which are violating the model assumptions corroborates (not failsafe, just an additional line of evidence) this statement from Schindler et al. (2017).

"Such increases in nighttime oxygen concentrations were observed in several of our study streams and appear to be diagnostic of two-stage ecosystem metabolism."

It doesn't prove it, it's just an additional line of evidence.

[Figure]

[Figure]

**Fig. 1.** Modelled DO Ebble 16th May

[Figure]

[Figure]

**Fig. 2.** Modelled DO Avon 16th May

[Figure]

**Fig. 3.** Effect of assumed drift (Ebble 16th May)

[Figure]

**Fig. 4.** Discharge for two rivers (Avon and Ebble)

---

## Author Response (AR1)

**Oxygen dynamics and evaluation of the single station diel oxygen model across contrasting geologies** (Simon J. Parker)

New figures:
Figure 2 recast in terms of percent saturation.
Figure 6 – panel C added.
Figure 7 – new figure showing model residuals.

Response to referee comments.

| | |
|---|---|
| This manuscript uses a unique approach to evaluate whether the assumption of a constant rate of ecosystem respiration is valid over a daily cycle as assumed in most aquatic ecosystem metabolism models. He evaluates whether the point at which the rate of change in oxygen concentration for a given day is equal to zero (i.e., $dO2/dt = 0$) provides information about the ratio of ER/k within and across stream types. He then argues that because this technique does not agree with results from the nighttime regression approach of Hornberger and Kelly (1975) that the assumption of a constant daily rate of ER is invalid. | |
| However, beyond that, it is not clear to me how this approach provides an estimate of diel change in R, as stated in the abstract and discussion? We know that ER changes over the course of a day in response to temperature (Holtgrieve et al. 2010) and carbon substrates (Schindler et al. 2017, Sadro et al. 2014) but you generate a single ratio using this approach, not explicit rates of ER. | In response to these comments, a paragraph has been added in the introduction (p.2, lines 13-24) |
| In addition, I'm not sure what the ratio (ER/k) really describes – how does this get you additional information that you don't get by fitting a metabolism model, since the ratio of R/k doesn't give you any information on their relative magnitudes. And, wouldn't you still face the issue of equifinality (many values of ER and k that could produce a given ratio)? | |
| Further, you discuss the importance of correcting ER and k for temperature, but then don't consider that in your estimation of their ratio – wouldn't the diel variation in temperature have a lot to do with when the point of $dO/dt = 0$ occurs as well? In addition, the temperature correction is different for the two, so the degree of daily temperature fluctuation could impact the resulting ratio. | I think this comment is concerned with kinetic effects of temperature on ER and k. I included an additional panel in Figure 6 which shows that temperature is not responsible for anomalies. Also additional text (p.6, lines 18 - 22) |

| | |
|---|---|
| Zero change in DO has an equal element of uncertainty to it (when does DO/dt = 0?) as does fitting a nighttime regression (i.e., where does night begin?) so I'm not sure what you gain through using this approach? In addition, using the nighttime regression technique is no longer the most common way of estimating reaeration rates because of some of the shortcomings you mention and cite. | Effect of selection of observations used for calibration of nighttime regression is shown in Figure 12. |
| In summary, I have concerns about the significance of these findings given the degree to which the field has moved on in terms of approaches to metabolism models (e.g., Appling et al. 2018, Song et al. 2016) and the ability to estimate and constrain k (Appling et al. 2018, Raymond et al. 2012). In addition, the results as presented spend a significant amount of time discussing the degree of coherence in diel oxygen patterns (e.g., timing of max O2), rather than comparing potential daily fluctuations or cross-system differences in important metabolic parameters. I recognize that assessing differences in the magnitude and timing of daily fluctuations has some meaning in terms of understanding the magnitude of processes, but this discussion would be more meaningful had those processes also been quantified. | |
| Specific comments: | |
| Page 2, Lines 3-4: "Primary production can be quantified by partitioning a single DO time series into its component fluxes, namely photosynthesis, ecosystem respiration and aeration" – perhaps you mean to say "ecosystem metabolism" | Changed accordingly. |
| Page 3, lines 3-5: What is the "pertinent process"? | Line changed to: "if the model structure adequately captures DO dynamics, " |
| Page 3, lines 19-20: "primarily groundwater fed". . . couldn't the point of zero change reflect the O2 concentration of groundwater input as well? | Comment added p.6, lines 27-31. |
| Page 4, lines 27-28 – wouldn't it be "normalized" not "detrended"? | Changed accordingly. |
| | |
| I'm sorry this review is delayed but the delay has resulted in clarification of a number of my questions which were picked up by the other referee. In particular, I now have a much better understanding of the purpose of the paper. This was not clear on the initial reading. | |
| I would therefore strongly recommend that the | The abstract has been changed to refer to |

| | |
|---|---|
| abstract be revised to better reflect the purpose of the paper. i.e. the point stressing that this paper is " about evaluating model structures, not quantifying parameters", needs to be right up front. | model validity and includes a sentence about when single stage R models are likely to be less valid. |
| It is also worth reflecting on George Box's pithy aphorism that all models are wrong but some are useful. This is highly relevant to this manuscript as it begs the question 'useful for what?' There are a number of generalizations and simplifications implicit in solving the 'simple' metabolism model of Odum into three components viz. (ecosystem) respiration, primary production and reaeration. The author points out the likely effects of autochthonous vs allochthonous carbon on respiration rates and how this is likely to be time dependent on a daily time frame. I totally agree with this. The key point for me though is 'does this matter?' What question is driving the use of stream metabolism in the first place? If it's mechanistic understanding, then nuances matter very much. If it's about aggregating organic carbon loads across time and space, contrasting watersheds with different land use for example, then it doesn't matter anywhere near as much, if at all. | |
| As a general point, I find discussion centered on changes in DO based on temperature often inadequate as the obvious effect of temperature effect on DO solubility is neglected. An increase in night time DO is expected if water temperature falls. Framing the discussion in terms of %DO saturation is much more useful to examine the interplay of R and k but in this instance makes it more convoluted to then talk about change in DO = 0. Of course if temperature doesn't change (nor atmospheric pressure or salinity to be pedantic) then change in DO = 0 would correspond to a change in %DO saturation of 0. | Figure 2 has been re-framed in terms of %DO saturation and comments about increasing %sat on p.4, lines 28-30.

Effect of falling water temperature, p.9, lines 19-21 |
| Minor point: Page 2, Line 4. Suggest changing 'components' as this word was used in the previous sentence to refer to P, R & reaeration. | Changed to 'parts'. |
| Because this is novel, I would like to see a little more explanation about HOW R/k can be used to interrogate the validity of the model used for fitting diel O2 curves. This may be immediately obvious to those who routinely inverse model metabolism, but to many readers this won't be clear at all. | Paragraph added in Discussion section (paragraph 2 of discussion). |
| Page 2, Line 20. Equifinality hinders resolving ER and k when %DO saturation is very close to 100% or when there is very little change in %DO over the day, it is not a universal problem. | I deleted the sentence regarding equifinality as I think it is a diversion. |
| Page 3, Line 26. It has already been stated that | Probe drift was analysed in discussion |

| | |
|---|---|
| precision and accuracy of the DO sonde data is of fundamental importance in reliably identifying points where the change in DO is zero. Yet there is no mention at all of how accuracy of the deployed sondes was verified. Probe drift would be a major confounding factor in this analysis. | paper (including model and figures). Comment added p.9, lines 31-32. |
| Page 5, Line 13. The cause of this sudden change in the rate of decline? | I don't know. I could speculate that it is because labile organics have been consumed, but that would be too convenient for the overall argument (although that is a possible explanation). It is a long time series (each river is about half million records), so it's hard to explain local features. |
| Is there any significant time-of-day dependent topographic shading of any of these streams over the study reach integrated by the sonde? If so, this may then affect time of peak DO. | Yes, this is true. This cannot be ruled out. Also, if time to peak is shorter duration, then time to minimum (after sunset) is likely (although not inevitably) to be shorter duration. But early time to peak (and early time to minimum) for both Nadder and Ebble together with the fact that it is those two which are violating the model assumptions corroborates (not failsafe, just an additional line of evidence) this statement from Schindler et al. (2017).

"Such increases in nighttime oxygen concentrations were observed in several of our study streams and appear to be diagnostic of two-stage ecosystem metabolism." |
| Reaeration will not only depend on temperature (in a well-known relationship) but also on wind (there is a lot of lakes' literature on this topic) and on discharge. A change in discharge will almost certainly change k and this relationship will be idiosyncratic for each site depending on stream channel shape, wetted area, roughness etc. Are these additional factors responsible for some of the variation observed in this data set? | Paragraph added in discussion p.9, lines 5-15

The important aspect is not whether k could change, but whether it could change over the course of a single night and follow the same pattern of change over several nights. If it just changes from day to day, that would not result in violation of model assumptions. |
| Bearing in mind the already posted review and the author's responses, I still believe this is an interesting paper that definitely warrants publication. However, to be more useful to the general reader and in particular those undertaking studies where stream metabolism is being measured, I strongly recommend the author provides an additional paragraph or two which guides the user through checking the inherent | Paragraph 2 in discussion. |

| model assumptions when modelling their data. This can be in a series of steps checking whether the assumption of an invariant R (temperature effects notwithstanding) has a significant effect on overall metabolic parameter estimates. | |
|---|---|

[revised manuscript text omitted]